# Methods of Groundwater Recharge Estimation under Climate Change: A Review

Riwaz Kumar Adhikari [1,*], Abdullah Gokhan Yilmaz [1], Bandita Mainali [2], Phil Dyson [3] and Monzur Alam Imteaz [4]

1   Department of Engineering, La Trobe University, Melbourne 3086, Australia
2   School of Engineering, Faculty of Science and Engineering, Macquarie University, Sydney 2109, Australia
3   North Central Catchment Management Authority, Huntly 3551, Australia
4   Department of Civil and Construction Engineering, Swinburne University of Technology, Melbourne 3122, Australia
*   Correspondence: r.adhikari@latrobe.edu.au

**Abstract:** Groundwater resources have deteriorated in many regions as a result of excessive use to satisfy increasing water demands. Furthermore, climate change has an influence on groundwater in terms of quality and quantity. An investigation of climate change impacts on quality and quantity of groundwater is vital for effective planning and sustainable management of groundwater resources. Despite of the importance of climate change impact studies on groundwater resources, climate change impact studies related to surface water resources have attracted more attention from the research community, leading to limited understanding of the groundwater and climate change relationship. In this paper, a systematic review of the latest literature related to the impact of climate change on groundwater recharge was carried out to provide guidance for future studies.

**Keywords:** climate change; groundwater; recharge; uncertainties; hydrological models

## 1. Introduction

Groundwater is a huge reserve of water underneath the Earth's surface vital for humans and ecosystems. Approximately a third of the water used originates from underground sources [1,2] and nearly 2 billion people globally utilize groundwater for drinking purposes [3]. In regions lacking sufficient surface water supply from rivers and reservoirs, groundwater is critical for meeting the water demand. The demand for groundwater is rapidly increasing with the rise in population. Furthermore, climate change is imposing additional stress on surface and groundwater sources. Therefore, it is essential to understand climate change impacts on groundwater resources to achieve sustainable water resource management [4].

Climate change influences groundwater systems in several ways. The most direct climate change effect on groundwater is related to the groundwater recharge as climate change can alter rainfall amounts and patterns, thus changing the quantities of soil infiltration, deep percolation, and hence the recharge. Moreover, a rise in temperature increases the rate of evaporation, reducing the quantity of water available to replenish groundwater. In addition to groundwater recharge effects, climate change has significant influences on groundwater quality. Climate change has an influence on groundwater quality through changes in physical, chemical, and biological properties of the aquifer in both drier and wetter scenarios, as these properties are highly related to climatic factors. Under drier conditions, an increase in total dissolved solids through an increase in salt content will cause groundwater quality degradation [5].

Regions highly dependent on groundwater can be greatly affected by climate change. Thus, studying the impacts of climate change on the quality and quantity of groundwater is of utmost importance in those regions for meeting future groundwater demands. This study

aims to provide a systematic review of the available literature for climate change effects on groundwater recharge globally to provide recommendations for future groundwater studies. Although some review studies on climate change and groundwater relationship have been conducted in the past [6,7], this study has included the latest climate change impact studies globally to cover the most recent methods in climate and groundwater modelling to better guide future groundwater studies.

## 2. Materials and Methods

A systematic review of the literature on the climate change and groundwater relationship was conducted in this study. Figure 1 shows a schematic flowchart of the conducted literature review.

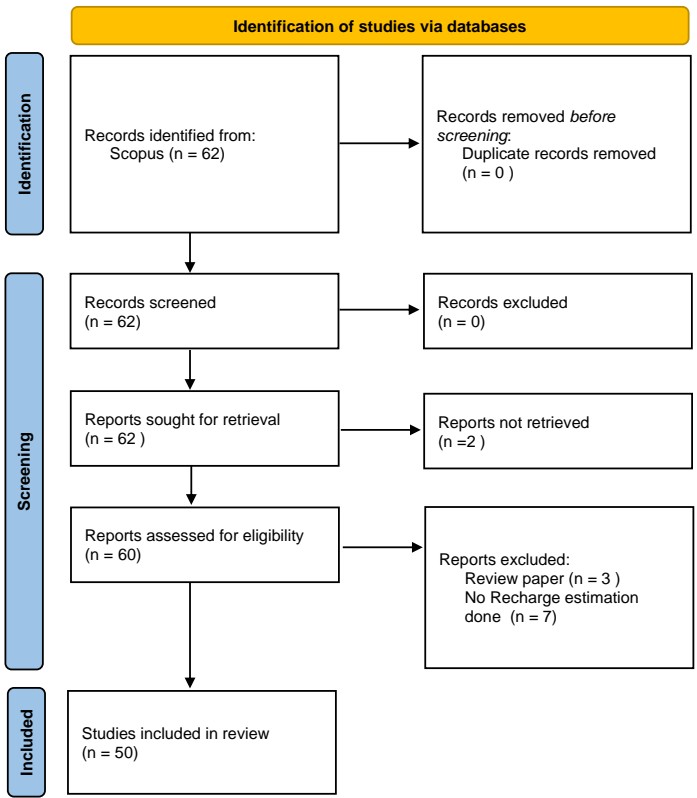

**Figure 1.** PRISMA search strategy used for the literature review.

The literature review was conducted for studies related to the impacts of climate change on groundwater recharge. The resource search was restricted to publications from peer-reviewed journals in English language. Conference proceedings were excluded. The most current studies over the period of 2016–2022 were considered to be reviewed. The study search was performed through the SCOPUS database using keywords "groundwater", "recharge", and "climate change". The schematic flowchart of the literature review using the PRISMA search strategy [8] is shown in Figure 1.

The initial search for climate change impacts on groundwater recharge using keywords "groundwater", "recharge", and "climate change" yielded 62 studies globally. Following the initial search, the studies were further eliminated based on relevance after reading the abstract and then the whole paper. Studies that conducted recharge estimation were considered in this paper. Moreover, review papers were removed from the list of selected studies. According to above explained methodology, 50 studies regarding the impacts of climate change on groundwater recharge in the literature were selected and reviewed in this study. The study areas (countries) of the reviewed studies are shown in Figure 2 and the publication years of the studies are shown in Figure 3 for the selected studies.

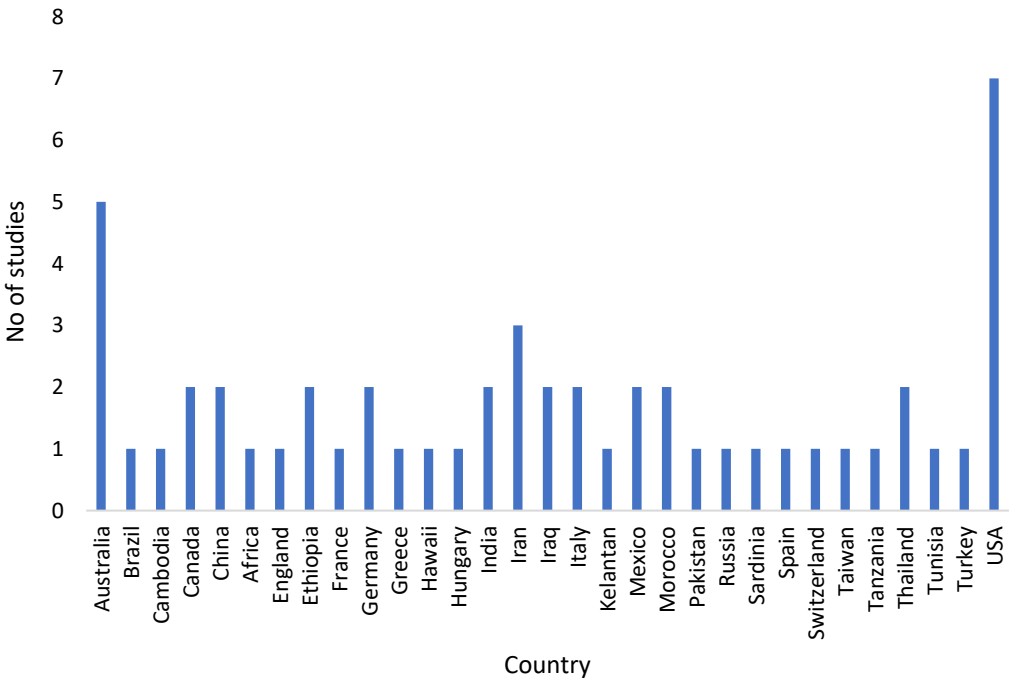

**Figure 2.** Study area of the review studies.

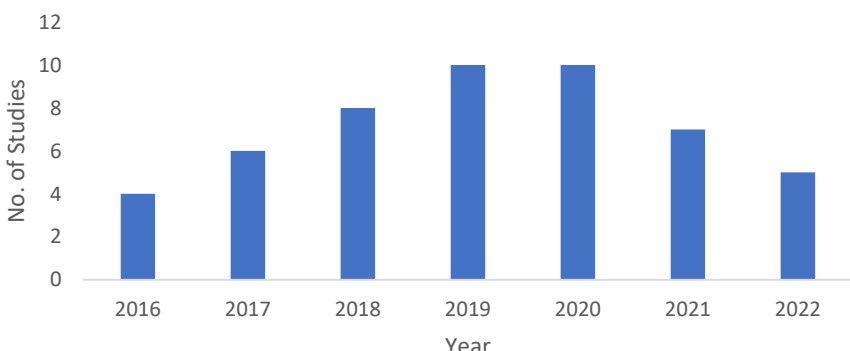

**Figure 3.** Year of publication of the selected studies.

It can be observed from Figure 2 that the majority of the selected studies have been conducted in United States, followed by Australia. Figure 3 indicates a growing interest in studies related to climate change impact on groundwater until 2020, followed by a decline in the number of studies in 2021, possibly related to the effects of COVID-19 on research globally.

## 3. Studies on Groundwater Quantity

Climate change impact on groundwater quantity studies can be grouped into two broad categories: (1) studies investigating climate change and variability impacts on groundwater through analysis of observed data sets, and (2) studies producing groundwater projections using climate projections in calibrated hydrological models.

### 3.1. Studies on Groundwater Quantity through Analysis of Observed Datasets

Observed data analysis is employed to establish climate variability and change impact on groundwater, as well as to identify the sensitive parameters that will further assist the projection studies. This is performed either through hydrological modelling or through statistical trend analysis techniques. Hydrological models are numerical modelling tools used for studying various hydrological processes. Trend analysis techniques are employed for identifying the existence or non-existence of significant trends in hydrological data. Trend analysis can be performed using parametric and/or non-parametric tests. Parametric tests are developed based

on the assumption of normality, stationarity, and the independency of time series. In cases where parametric tests are not appropriate, non-parametric tests are used. Non-parametric tests employ some means to rank the measurements and test for weirdness of the distribution. When the distribution is skewed (not normal), or unknown, or the sample size is too small to presume a normal distribution, non-parametric tests are necessary. Likewise, non-parametric tests are useful when there is the existence of outliers in the dataset [9]. Therefore, non-parametric methods are employed for hydrological time series analysis [10].

A few of the selected studies examined the impact of climate change on groundwater recharge through analysis of the historical observed data using hydrological models. The authors of [11] assessed the impact of change in land use and climate variability on recharge in the upper Gibe watershed from 1985 to 2018, and reported a significant decrease in recharge with regards to land use/cover change and climate variability using a distributed hydrological model, WetSpass. A reduction in groundwater recharge in southern Germany was reported by [11] through simulation of recharge from 1951 to 2019 using a soil water balance model. The authors of [12] studied water budget and groundwater recharge using HYDRUS 1-D model in Russia. The study showed that, despite a significant increase in air temperature, groundwater recharge did not change in the southern regions in Russia, while increments of up to 50–60 mm/year in recharge in the central and northern regions of Eastern Russia were observed.

Some studies reviewed in this paper have performed historical analysis of the observed datasets using statistical trend analysis techniques. The authors of [13] studied spatio-temporal variability of groundwater level in South-West Western Australia (SWWA) using non-parametric statistical tests, namely the Mann–Kendall test and Kolmogorov–Smirnov test. A d reduction in groundwater level by 13 mm per month in the coastal region of SWWA was found, which was concluded to be caused by anthropogenic impacts as the primary factor and climate variability and change as the secondary factor. Several statistical methods were used in investigating drivers (both climatic and non-climatic) behind variations in recharge using 43 years of data from 426 bores in South-East South Australia by [14]. Pearson's correlation coefficient and least-squares linear regression were used to examine the potential relations between annual groundwater recharge, calculated using the water-table fluctuation method, and the considered drivers. The authors of [15] applied linear regression, multi-layer perception (MLP) and long short-term memory (LSTM) models to forecast recharge in South Australia and Victoria, Australia.

### 3.2. Studies on Groundwater Projection

Groundwater recharge projections are important in order to quantify the probable impacts of climate change on groundwater resource, so that important interventions can be made. However, recharge is the most challenging component of the water balance to estimate as it cannot be directly measured. Recharge is influenced by complex natural processes, both spatially and temporally, making its measurement and future estimation very challenging. Recharge estimates are carried out mainly through chemical tracers [16–18] and hydrological modelling approaches [19,20].

Hydrological models use water balance estimation to calculate various components of water balance. They are commonly used for future projection of groundwater recharge. The groundwater recharge projection process starts with choosing a global climate model (GCM) or regional climate model (RCM) or sets of GCMs and RCMs to produce (future) climate projections under one or more greenhouse gas emission scenarios. The spatial resolution of the GCM outputs is coarse and needs to be further downscaled to a finer resolution suitable for application in hydrological modelling studies. The GCM outputs, after downscaling, are then applied with calibrated hydrological models to simulate specific groundwater component, mostly recharge. The whole process including GCMs' selection, downscaling methods, and hydrological models creates uncertainties in outcomes. A review study by [21] identified that the prediction of future recharge is affected by the choice of climate model, downscaling technique, recharge model, and emission scenario. Land use considerations are another significant source of uncertainty, which should be in line with the emission scenarios. The

identification of uncertainties in impact studies and its quantification indicate the level of confidence in the results of studies. Although uncertainties are inherently a part of impact prediction studies, proper selection of the methodology can help the results become more applicable. Thus, an attempt was made to categorize the projection studies based on five major sources of uncertainties (i.e., (1) downscaling techniques and number of GCMs used, (2) scale of study, (3) modelling approach, (4) land use change considerations, and (5) uncertainty considerations) to identify the best method to minimize the uncertainties.

### 3.2.1. Downscaling Techniques and Number of GCMs Used

It is essential to produce projected (future) climate data first to be able to generate groundwater projections using future climate data in calibrated groundwater models. GCMs' outputs are traditionally used to produce climate projections. GCMs are mathematical models for representing the physical processes in the atmosphere, ocean, cryosphere, and land. They are the only source to simulate and predict time series of climate variables on a global scale. This is carried out by dividing the globe into horizontal grids (250 km to 600 km) with vertical layers (10–30). They simulate the change in climatic variables such as precipitation, temperature, humidity, and wind speed [22].

As mentioned before, the spatial resolution of GCMs is too coarse; therefore, the GCM outputs should be downscaled into a finer resolution to better account for regional climatic influences. There are two main approaches to downscale GCMs' outputs: statistical and dynamic downscaling. Dynamical scaling is conducted through the use of higher resolution climate models called regional climate models (RCMs). The mathematical structure of RCMs is similar to GCMs; however, RCMs result in higher resolution output in comparison with GCMs, as RCMs focus on a limited area of interest. The climate variables derived by the GCMs are used as an input for the RCMs. If RCM data are not available for the study area or its output is still too coarse, the statistical downscaling approach is employed to generate high-resolution data. In statistical downscaling, a statistical relationship is developed between the historic observed climate data and the climate model output for the same historical period [23]. The relationship is then used to develop the future climate data. The majority of the reviewed studies (88%) have adopted statistical downscaling as it is easy to apply and also a cost-effective approach in comparison with dynamical downscaling. Only 12% of reviewed papers adopted dynamically downscaled data in their studies.

GCM and RCM simulations vary from observed climate as a result of systematic and random model errors known as biases, which inherently exist during downscaling and require post-processing, called bias correction, before they can be applied. There are various types of bias correction techniques such as the delta change approach, linear scaling method, multiple linear regression, power transformation, analogue method, local intensity scaling, and quantile mapping. Bias correction was conducted in 97% of the reviewed studies.

A combination of multiple GCMs' output was used in 94% of the studies reviewed in this paper, and more than one greenhouse gas emission scenario was adopted in all reviewed papers. For example, the authors of [24] used data from three GCMs including CanESM5, EC_Earth3, and MIROC6 out of seven through performance evaluation using the entropy method. Linear bias correction of the GCM outputs was performed for two shared socioeconomic pathways (SSPs)—SSP2-4.5 and SSP5-8.5. Three GCMs under two representative concentration pathways (RCP 4.5 and RCP 8.5) were used to project future rainfall and temperature in a study in Pakistan by [19]. When multiple climate models' outputs are used, an ensemble approach is commonly adopted. The ensemble evaluates the results from multiple GCMs' output for the same variable using the mean or median. For example, the authors of [25] assessed the impact of land use and climate change in East Africa using an ensemble of GCMs under RCP 4.5 and RCP8.5. The authors of [26] investigated the impact of future climate on groundwater resources of British mainlands using an ensemble of 11 models for rainfall and evaporation. The GCMs and RCMs used in the reviewed studies with the time step, spatial resolution, and bias correction techniques are summarized in Table 1.

**Table 1.** Details of the climate models used in the study.

| Study Number | Title of the Study | GCM | RCM | Time Step | Spatial Resolution | Downscaling Technique | Bias Correction |
|---|---|---|---|---|---|---|---|
| 1 | Impacts of climate and land use change on groundwater recharge under shared socioeconomic pathways: A case of Siem Reap, Cambodia [24] | BCC-CSM2-MR, CanESM5, EC-Earth3, GFDL-CM4, IPSL-CM6A-LR, MIROC6 CSSR, MRI-ESM2-0 | NA | Monthly | 1.13°, 2.81°, 0.7°, 1°, 1.98°, 1.41°, 1.13° | NA | Linear scaling method |
| 2 | Impacts of climate and land-use change on groundwater recharge in the semi-arid lower Ravi River basin, Pakistan [19] | 1. CCSM4 2. MPI 3. MIROC5 | NA | Daily | 0.44° | Quantile mapping | Quantile mapping |
| 3 | Impact of climate change on groundwater recharge in the lake Manyara catchment, Tanzania [20] | 1. MPI- ICHEC CNRM 2. ICHEC 3. MPI-ICHEC CNRM 4. ICHEC | 1. CLM com COSMO –CLM (CCLM4) 2. DMI HIRHAMS 3. SMHI RCA4) 4. KNMI | Daily | 0.5° | Dynamical downscaling | NA |
| 4 | The effect of climate change on groundwater recharge in unconfined aquifers in the western desert of Iraq [27] | CanESM2 | NA | Daily | 2.8125° | Statistical downscaling | NA |
| 5 | Climate Change Impacts on Groundwater Recharge in Cold and Humid Climates: Controlling Processes and Thresholds [28] | ACCESS1-0_rcp45_r1i1p1, ACCESS1-3_rcp85_r1i1p1, bcc-csm1-1-m_rcp45_r1i1p1, BNU-ESM_rcp85_r1i1p1, CanESM2_rcp45_r1i1p1, CMCC-CMS_rcp45_r1i1p1, GFDL-CM3_rcp45_r1i1p1, GISS-E2-R_rcp45_r6i1p3, inmcm4_rcp45_r1i1p1, MIROC-ESM _rcp45_r1i1p1, MIROC-ESMCHEM_rcp85_r1i1p1, MRI-ESM1_rcp85_r1i1p1 | NA | Monthly | 10 km | Quantile mapping | Quantile mapping |

**Table 1.** *Cont.*

| Study Number | Title of the Study | GCM | RCM | Time Step | Spatial Resolution | Downscaling Technique | Bias Correction |
|---|---|---|---|---|---|---|---|
| 6 | Climate and land-use change impacts on spatiotemporal variations in groundwater recharge: A case study of the Bangkok Area, Thailand [29] | NA | 1. ACCESSCSIRO-CCAM, 2. MPI-ESM-LR-CSIRO-CCAM, and 3. CNRM-CM5-CSIRO-CCAM | Daily | $0.5°$ | Quantile mapping | Quantile mapping |
| 7 | Assessing the Effect of Land/Use Land Cover and Climate Change on Water Yield and Groundwater Recharge in East African Rift Valley using Integrated Model [25] | bcc-csm1-1-m, MRI-CGCM3, and CMCC-CM | NA | Daily | $2.7906° \times 2.8125°$, $1.12148° \times 1.125°$, $0.7484° \times 0.75°$ | Quantile mapping | NA |
| 8 | The impact of climate change on groundwater recharge: National-scale assessment for the British mainland [26] | HadRM3-PPE | NA | Daily | 1 km | NA | NA |
| 9 | From Flood to Drip Irrigation Under Climate Change: Impacts on Evapotranspiration and Groundwater Recharge in the Mediterranean Region of Valencia (Spain) [30] | ICHEC-EC-EARTH—CCLM4-8-17, ICHEC-EC-EARTH—HIRHAM5, MPI-M-MPI-ESM-LR—CCLM4-8-17, CNRM-CM5—CCLM4-8-17, CNRM-CM5—ALADIN63 | NA | Daily | $0.11°$ | NA | Quantile mapping |
| 10 | Evaluation of climate change impact on groundwater recharge in groundwater regions in Taiwan [31] | CMIP3 | NA | Monthly | 25 km | NA | NA |
| 11 | Impact of climate change on groundwater recharge in a Brazilian Savannah watershed [32] | HadGEM2-ES and MIROC5 | Eta-HadGEM2-ES and Eta-MIROC5 | Daily | 20 km | NA | Linear bias correction |
| 12 | Proportional variation of potential groundwater recharge as a result of climate change and land-use: A study case in Mexico [33] | MPI-ESM | NA | Monthly | $1.9°$ | NA | NA |

**Table 1.** *Cont.*

| Study Number | Title of the Study | GCM | RCM | Time Step | Spatial Resolution | Downscaling Technique | Bias Correction |
|---|---|---|---|---|---|---|---|
| 13 | Climate change impact on surface water and groundwater recharge in northern Thailand [34] | 1. MIROC5 2. MPI–ESM–MR 3. CNRM–CM5 | NA | Daily | 1° | NA | Change factor method |
| 14 | A water balance model to estimate climate change impact on groundwater recharge in Yucatan Peninsula, Mexico [35] | 1. CNRM-CM5, 2. GFDL_CM3, 3. HADGEM2-ES, 4. MPI_ESM_LR | NA | Monthly | 0.5° | Change factor method | NA |
| 15 | Climate change impact on groundwater recharge of Umm er Radhuma unconfined aquifer Western Desert, Iraq [36] | HadCM3 | NA | Daily | 2.5° × 3.75° | Statistical downscaling | NA |
| 16 | Prediction of the response of groundwater recharge to climate changes in Heihe River basin, China [37] | HadCM3 | NA | Daily | 2.5° × 3.75° | Statistical downscaling | NA |
| 17 | Long-term effect of climate change on groundwater recharge in the Grand East region of France [38] | CanESM2, CCSM4, INM-CM4, ACCESS1.0, HadGEM2-ES, MRI-CGCM3, IPSL-CM5A-MR, CNRM-CM5, MIROC-ESM, MIROC5, CSIRO Mk 3.6, CESM1-CAM5, MPI-ESM-LR, GFDLCM3 and GISS-E2R | NA | Daily | 0.5° | Change factor method | NA |
| 18 | Climate change projections in the Ghis-Nekkor region of Morocco and potential impact on groundwater recharge [39] | NA | HIRHAM5, RACMO22T, and RCA4 | Daily | 0.44° | NA | Linear regression |
| 19 | Impacts of climate change on groundwater recharge in Wyoming big sagebrush ecosystems are contingent on elevation [40] | CMIP5 | NA | Daily | 0.5° | NA | NA |

**Table 1.** *Cont.*

| Study Number | Title of the Study | GCM | RCM | Time Step | Spatial Resolution | Downscaling Technique | Bias Correction |
|---|---|---|---|---|---|---|---|
| 20 | Sensitivity of potential groundwater recharge to projected climate change scenarios: A site-specific study in the Nebraska Sand Hills, USA [41] | BCC_CSM, CANESM, CCSM, CESM_BGC, CNRM_CM, CSIRO, GFDL_CM, GFDL_G1, GFDL_M1, INMCM, IPSL_CM, IPSL_MR, MIROC_ESM, MIRCO_ESM, MICROC, MPI_LR, MPI_MR, MRI_GCM, NORESM | NA | Daily | 1°–3.75° | NA | NA |
| 21 | The effects of climate change on groundwater recharge for different soil types of the west shore of Lake Urmia—Iran [42] | HadCM3, CanESM2 | NA | Daily | 2.5° × 3.75°, 2.8125° | Statistical downscaling method | NA |
| 22 | Regional variations in potential groundwater recharge from spring barley crop fields in the UK under projected climate change [43] | NA | NA | Daily | 5 km | NA | NA |
| 23 | Predicting the impacts of climate change on groundwater recharge in an arid environment using modeling approach [44] | HADCM3 | NA | Daily | 2.5° × 3.75° | Dynamic downscaling | NA |
| 24 | Future irrigation expansion outweigh groundwater recharge gains from climate change in semi-arid India [45] | CCSM4, GFDL-ESM, HadGEM2-ESM, MIROC5, MPI-ESM | NA | Daily | 1.9°–2.5° | Modified delta method | |
| 25 | Irrigated agriculture and future climate change effects on groundwater recharge, northern High Plains aquifer, USA [46] | BCCR-BCM2.0, CGCM3.1(T63), CNRM-CM3, CSIRO-Mk3.0, GFDL-CM2.0, GFDL-CM2.1, GISS-ER, INGV-SXG, INM-CM3.0, IPSL-CM4, MIROC3.2, ECHAM5/MPI-OM, MRI-CGCM2.3.2, PCM | NA | Daily | 1°–3.75° | NA | NA |

Table 1. *Cont.*

| Study Number | Title of the Study | GCM | RCM | Time Step | Spatial Resolution | Downscaling Technique | Bias Correction |
|---|---|---|---|---|---|---|---|
| 26 | Impact of climate change on groundwater recharge and base flow in the sub-catchment of Tekeze basin, Ethiopia [47] | NA | l REMO | Daily | 55 km (0.44° × 0.44°) | NA | Linear bias correction method |
| 27 | Developing empirical monthly groundwater recharge equations based on modeling and remote sensing data—Modeling future groundwater recharge to predict potential climate change impacts [18] | CCMS3, CNRM, ECHAM5-r3, HADCM3-Q0 and IPSL | RCA3 | Monthly | 50 km | Dynamic downscaling | Distribution based scaling (DBS) |
| 28 | Changes in groundwater recharge under projected climate in the upper Colorado River basin [48] | CMIP5 | NA | Monthly | 0.125° | Statistical downscaling method | NA |
| 29 | Sensitivity of mGROWA-simulated groundwater recharge to changes in soil and land use parameters in a Mediterranean environment and conclusions in view of ensemble-based climate impact simulations [49] | HadCM3, ECHAM5 | RCA, REMO, RACMO2 | Daily | 25 km | Multi-fractal technique | NA |
| 30 | Climate change impact assessment on groundwater recharge of the upper tiber basin (central Italy) [50] | NA | RegCM, PROMES, RCAO | Daily | 50 km | NA | Delta change method |

As shown in Table 1, most of the studies were conducted using daily data with varied spatial resolution ranging from 0.44° to 3.75°. Statistical downscaling was the most commonly adopted downscaling technique. It is observed that some of the studies used the same method for both downscaling and bias correction technique (e.g., change factor method). Various climate models were used depending upon the location of the study. Climate model projection from secondary sources, which are already downscaled and bias corrected, were also used in some of the studies.

### 3.2.2. Scale of the Study

The impacts of climate change on groundwater are studied in one of four spatial scales: global, regional, basin, and aquifer levels. Global-scale studies evaluate the global pattern of projected trends in groundwater recharge and variability, providing an overview of prevailing conditions. However, these studies are too general to guide water policy and decision-making needed at smaller scales [51–53]. Basin or aquifer specific studies provide a detailed understanding of the effects of climate change in a particular catchment area or aquifer. According to [52], regional studies are useful to compromise between both scales as these evaluate a group of aquifers within a region, with similar or different recharge mechanisms. The majority of the reviewed studies are performed on a basin scale (54%), while 32% of the studies are on a regional scale. Only 14% of the reviewed studies are conducted on an aquifer scale.

### 3.2.3. Modelling Approach Used

Groundwater recharge estimation is a complex task as it cannot be directly measured, unlike other components of the water balance. As mentioned earlier in the paper, recharge is estimated through chemical tracers and hydrological modelling approaches. The chloride mass-balance method (CMB) is a widely used chemical tracer method [54]. Although the chemical tracer method was used by some studies in the literature [17,46], the hydrological modelling approach is more commonly employed for recharge simulation and estimation.

Hydrological models are standard tools employed for studying hydrological process and have various applications that range from small basins to global scale models. Every model has its own unique application, features, advantages, and disadvantages. There are several hydrological model classifications based on the input to the model, the parameters, and the degree of physical principles applied. Hydrological models can be classified as lumped and distributed according to model parameters as a function of space and time [55]. In lumped models, the entire catchment is taken as a single unit, while in distributed models, the catchment is divided into smaller units considering the spatial variability of input, parameters, and output. The majority of the papers reviewed in this study have used distributed models (66%), while the remaining (34%) have used lumped models.

Hydrological models are also classified as empirical, conceptual, and physically based models. Empirical models are data-driven models, which are observation-oriented and do not consider the various physical processes in a hydrological system. Unit hydrographs, statistically based models, and artificial-intelligence-based models are examples of empirical models. About 14% of the reviewed papers in this study used empirical models. The authors of [56] used statistical analysis to estimate recharge in Northern China. Spatio-temporal variability of groundwater recharge due to climate change in SWWA was studied using an empirical model by [13]. Conceptual models are based on modelling of interconnected reservoirs and involve semi-empirical equations with a physical basis [55]. The parameters of these models are derived from the field observations and through calibration. Physically based models are mathematically idealized representations of the physical phenomenon and processes and are popular because of the use of parameters with a physical interpretation [55]. About 33% of the studies reviewed used a conceptual model, while the majority (53%) of the studies used physically based models. The details of the models used in the reviewed studies are shown in Table 2.

**Table 2.** Details of the models used in the studies.

| | Model | Description | Input | Output | Pros | | Cons | |
|---|---|---|---|---|---|---|---|---|
| 1 | SWAT [18,19,24,25,37,50,57] | Physically based semi-distributed model | Basin characteristics (elevation, land use, soil) and Hydrometeorological data (temperature, precipitation, relative humidity, solar radiation, wind speed, discharge) | All water balance components—evapotranspiration, surface runoff, and recharge | 1. 2. 3. 4. 5. | Long-term prediction of climate change impact is possible Flexibility in the catchment size, can be used for small and large catchments Land use change effect can be modelled Ability to describe sediment transport, vegetation growth, and nutrient transport Freely available | 1. 2. | Dynamic land use not possible and dew point not considered Complex—requires human expertise and computational capability |
| 2 | WetSpass [11,17,20,27,29,36,47] | Physically based distributed model | Biophysical characteristics (elevation, land use, soil) and seasonal hydro-meteorological data—(temperature, precipitation, potential evapotranspiration, wind speed, groundwater level) | Spatial distributed groundwater recharge, infiltration rates, soil moisture, and runoff. | 1. 2. 3. | Ability to simulate groundwater recharge in areas with limited observed data availability Long-term prediction of climate impact is possible Freely available | 1. 2. | Dynamic land use not possible Complex and requires human expertise and computational capability |
| 3 | ZOODRM [26] | Conceptual distributed recharge model | Basin characteristics—land use, soil type, topography (DEM), geology along with a river network and potential evapotranspiration, moisture content at field capacity and permanent wilting point | Runoff and recharge in nodes | 1. 2. | Considers soil moisture deficit Simple model | 1. 2. | Potential recharge is calculated, not the actual recharge Large data requirement |
| 4 | HBV [31] | Semi distributed conceptual model | Hydrometeorological data and other field-measured parameters | All water balance components—recharge, runoff, and actual evaporation | 1. 2. | Simple model implemented on computer code Parameters derived from field data and calibration | 1. 2. 3. | Large data requirement Calibration involves curve fittings, which make physical interpretation difficult Land use change effect cannot be modelled |
| 5 | Hydrus-1D [12,41,46,58] | Conceptual model | Hydrometeorological data and other field-measured parameters | All water balance components. | 1. | Powerful in modelling spatiotemporal ionic strength of soil | 1. | Data intensive |

**Table 2.** *Cont.*

| | Model | Description | Input | Output | Pros | Cons |
|---|---|---|---|---|---|---|
| 6 | MIKE SHE/MIKE11 [45] | Physically based distributed model | Basin characteristics and hydrometeorological data | All water balance components. | 1. Long-term prediction of climate change impact is possible<br>2. Flexibility in the catchment size, can be used for small and large catchments<br>3. Land use change effect can be studied<br>4. Ability to account for sediment transport, vegetation growth, nutrient transport, river flow, and groundwater flow (saturated and unsaturated) | 1. Complex and requires human expertise and computational capability<br>2. Not freely available |
| 7 | MODFLOW [25,34,44,58] | Physically based numerical groundwater flow model | Recharge, hydraulic parameters, well initial heads and stream flows details | Transient groundwater elevations, surface water flows, elevations, and groundwater interactions in modelled streams | 1. The steady and unsteady flow in confined and unconfined aquifers can be simulated considering the effects of wells, rivers, drains, head-dependent boundaries, recharge, and evapotranspiration | 1. Flow velocity is not accurately quantified<br>2. Transient studies require data in same time frame, which can be troublesome |
| 8 | SWB (Soil Water Balance model) [48] | Distributed Conceptual model | Land cover, soil properties, and daily meteorological data | Temporally and spatially variable gridded estimates of potential recharge | 1. Specific to groundwater recharge estimation<br>2. Simple | 1. Other water balance components are not calculated<br>2. Large requirement of data |
| 9 | Hydro-Budget [28,59] | Distributed Conceptual model | Meteorological data (daily precipitation and temperature) and spatially distributed basin data (pedology, land use, and slopes) | Spatially distributed estimates of groundwater recharge | 1. Designed as an accessible and computationally affordable model<br>2. Long-term simulation for recharge | 1. Other water balance components are not calculated<br>2. Huge requirement of data |

Hydrological models are effective tools used for sustainable water management. The identification of model parameters is a challenging task owing to the simplification of complex natural processes, high spatial and temporal variability, and limited observations. Moreover, physically based models require parameter adjustments owing to discrepancies in observed values and the scale of modelling. Conceptual models require model calibration to estimate parameter values [60]. Calibration and validation are important steps to decide on the capability of the hydrological models. The measured component of the hydrological cycle is used for calibration and, in most of the cases observed, river discharge is used for calibration. In the reviewed studies in this paper, 46% of the studies used measured (observed) discharge for calibration. A manually calculated water balance component, either evapotranspiration, recharge, or discharge, is used for calibration in the remaining 54% of the studies.

Physically based models including SWAT and WetSpass are the most commonly used hydrological models for future projection in the reviewed studies. This is because of the capability of these models for accurate long-term predictions, flexibility in the size of the catchment, and ability for the incorporation of land use change. It is worth noting that coupled hydrological and groundwater models are recommended as they remove the disparity in capturing the spatial extent, improve the overall water balance estimation, and reduce computational burden. For example, the authors of [61] concluded that the coupled hydrological and groundwater model can produce comparably improved simulations of low flows in the stream network improving the water balance estimation.

### 3.2.4. Land Use Change Consideration

Temporary (vegetation change) or permanent (built-up area expansion) land use change has a significant impact on the groundwater recharge through modification of water balance processes. Several studies related to land use change on groundwater recharge have been conducted. For example, the authors of [62] reported a maximum of 52% reduction in future recharge due to the increment of built-up area in Ho Chi Minh City, Vietnam. Recharge was estimated to be reduced by up to 38% as a result of land use change in the lower Ravi river basin, Pakistan by [19]. Although land use change has a significant impact on the groundwater recharge, only 26% of the reviewed studies in this paper considered land use change projection along with climate change projection. For example, the authors of [24] studied the impact of climate and land use change on groundwater recharge in Cambodia. They generated land use projections through the Dyna-CLUE model. Similarly, the authors of [19,29] performed studies with the consideration of the land use projection by forming various scenarios using Dyna-CLUE. Some other models, namely CA-Markov and ANN-based cellular automata, have also been used for land use change projection studies. The authors of [25] used ANN-based cellular automata for land use projection in a study carried out in Africa to assess the combined effect of land use and climate change impact on groundwater recharge. The authors of [19] used the CA-Markov model for land use projection in a groundwater recharge study in the Ravi River basin in Pakistan. The consideration of land use change in terms of built-up area increment has been studied in most of the studies; however, agriculture practice changes were not taken into account in many studies. In agricultural intense regions, a change in crop can have significant impact on water available for recharge. As an example of studies considering agricultural practice change (land use change), the authors of [45] studied the impact of climate and irrigation practice change in an agricultural basin in India and reported that future irrigation expansion would result in the drying of wells, despite gaining water as a result of climate change.

### 3.2.5. Uncertainty Considerations

The quantification, determination, and analysis of uncertainty is an essential part of environmental and hydrological studies. There are several uncertainties associated with the projection of future groundwater recharge under a changing climate. The major uncertainty

in the projection studies is due to the resolution of climate models and the downscaling method. Moreover, the uncertainty associated with the emission scenario consideration and the limitation of the modelling process should be considered in groundwater recharge projection studies. As explained in [63], two metrics have been recommended to examine the sources of uncertainty: (1) qualitative reporting including degree of confidence and evidence and (2) quantitative measurements.

Uncertainty analysis has been performed in 28% of the reviewed studies. Only qualitative analysis, in which the sources of uncertainty have been discussed, was conducted in those papers. For example, the authors of [26] discussed the sources of uncertainty associated with climate models and the hydrological models used in the national-scale assessment of the British Mainland. The authors of [30] evaluated the uncertainty and reported in agreement with other studies that the largest sources of uncertainty in hydrological projections are typically from climate models, followed by the downscaling method and the choice of the hydrological model structure and parameterization. The authors of [17] identified the hydrological model as an important source of uncertainty in the study carried out in El Alem Nadhour Saouaf basin in Tunisia.

The most direct impact of climate change on groundwater is related to recharge, and this paper's focus is on the climate change effects on groundwater recharge. However, it is worth noting that the water quality of the aquifers is also impacted by climate change, with an influence on the physical, chemical, and biological properties of the aquifer. Limited research has been carried out and not much is known about the influence on groundwater quality by climate change. Groundwater quality degradation can occur through a variety of point or non-point sources such as agricultural practices, industrial wastes, and pharmaceutical wastes. This degradation can be exacerbated by climate change. An integrated approach is required to understand the impact on groundwater quality.

As an example of climate change impact studies on groundwater quality, the authors of [64] assessed annual behaviour and trends in dissolved ions from the years 2004 to 2020 in three springs of the Sibillini aquifer, Italy. Moreover, cross-correlation analysis was used to assess the impact of climate variability on spring water quality and discharge. A variation in groundwater chemistry, especially in calcium and sulphates, was observed in rainier years.

An integrated approach for the assessment of the impact of climate change along with land use change on groundwater quality and quantity provides a better understanding for policy makers. This integrated approach was conducted in a few reviewed studies. The authors of [65] studied the combined impact of climate change and land use change on groundwater quality and quantity in Spain. In this study, nitrate concentration was modelled along with the coupled hydrological and groundwater model using data from three GCMs. It was observed that nitrate concentration increased in all scenarios of climate and land use change considered in the study. It was recommended to reduce fertilizer applications along with better management practices to reduce nitrogen concentration in the groundwater system in the aquifer. The authors of [66] employed a coupled groundwater model with the surface water model using dynamically downscaled future climate data to investigate climate change effects on the water quantity and quality of a lake in Kalundborg, Denmark. The results of the study indicate that the nutrient load to the nearby water bodies including the lake is likely to increase significantly owing to effects of climate change. Climate change can also have indirect effects on groundwater quality as a result of climate-change-induced events such as droughts. As an example, the authors of [67] found that intensified drought events (due to climate change) in Botswana caused water shortages, which affect sanitation behaviour, increasing pit latrine use and causing an increase in nitrate, coliform, and caffeine concentrations in groundwater.

## 4. Conclusions and Recommendations

Groundwater resources in many regions have deteriorated as a result of excessive use of groundwater to satisfy increasing water demands. In addition, climate change affects

groundwater in terms of both quality and quantity. Investigation into climate change impacts on groundwater is essential for effective planning and sustainable management of water resources. Climate change impact studies will assist in better understanding of the groundwater system, which will provide a basis for future interventions. Despite continuous improvement in methods, climate change impact studies are not commonly used for decision making owing to inherent uncertainties associated with such studies. Moreover, assessing the climate and groundwater relationship is challenging as groundwater response to climate forcing is slow, unlike surface water systems.

In this paper, 50 studies related to climate change effects on groundwater were selected and reviewed through a systematic framework to guide future climate change impact studies. The focus of this review paper is climate change effects on groundwater quantity (recharge); however, a limited number of papers about climate change and groundwater quality relationship were also reviewed. The following are the main conclusions and recommendations from this review study:

- Using an ensemble approach to reduce uncertainty: There are several sources of uncertainties associated with climate change impact projection studies. The projections from GCMs and RCMs are the largest source of uncertainty in climate change impact studies. Several reviewed studies (e.g., [25,26,28,29]) recommended the use of multiple climate models' output, named the "ensemble approach", to conduct such studies. It is also recommended by several studies (e.g., [24,68,69]) to assess the performance of the climate models through the climate model selection process before employing their output in a climate change impact study.

- Proper downscaling method selection: A proper downscaling technique can minimize the errors associated with the process. The use of the statistical downscaling approach is recommended in comparison with the dynamical downscaling approach.

- Inclusion of land use projections in the impact study: It is recommended to adopt land use change scenarios along with climate change projections in groundwater impact studies to produce more applicable results and for water managers or other stakeholders to make more informed decisions. Moreover, it is advised to consider both temporary (vegetation change) and permanent (built-up area expansion) land use change projections. Moreover, the emission scenario and land use change scenario should be in line with each other. Multidisciplinary research combining social science and economics with land use projection studies will result in more comprehensive decisions.

- Selection of hydrological model(s)**:** The selection of hydrological models has a great influence on the study. The hydrological model to be used should be flexible in terms of the size of the catchment to be studied and be able to incorporate land use change. Physically based models have long-term prediction capabilities and flexibility in size of the catchment to be modelled and can incorporate land use change. Moreover, distributed models capture the spatio-temporal variation of the impact of climate change on the groundwater system. Thus, physically based distributed hydrological models are recommended for impact projection studies.

- Quantification of uncertainty: Uncertainties of the impact studies should be quantified, not just assessed qualitatively. The level of confidence in future recharge projections can be evaluated through considering, quantifying, and stating all possible sources of uncertainty. This will help the decision makers, water managers, and stakeholders to make decisions for better groundwater management.

- Groundwater quality assessment: Studies about climate change effects on groundwater quality are limited. An integrated approach considering the impact on groundwater quality and quantity is recommended for future studies.

**Author Contributions:** R.K.A. and A.G.Y. conducted the literature review; R.K.A., A.G.Y., B.M. and M.A.I. conceptualized the paper; R.K.A. prepared the draft manuscript; A.G.Y., B.M., P.D. and M.A.I. reviewed the paper. All authors have read and agreed to the published version of the manuscript.

**Funding:** This research received no external funding.

**Institutional Review Board Statement:** Not applicable.

**Informed Consent Statement:** Not applicable.

**Conflicts of Interest:** The authors declare no conflict of interest.

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
