# Peer review of "Methods of Groundwater Recharge Estimation under Climate Change: A Review"

_sustainability, doi:10.3390/su142315619_

Round 1

Reviewer 1 Report

Authors have done a review for Groundwater recharge estimation under climate change. Some climate and hydrological models were summarized in tables. However, the discussion is too superficial. It is not recommended to publish a review with simple statistical figure and tables. A good literature review should  include meta analysis and bibliometrics.  I would be pleased to review the revised manuscript with these analyses.

Reviewer 2 Report

This manuscript is a review article on the impact of climate change on groundwater, which has adequately investigated and described existing studies. It has researched many papers on groundwater research, and has been appropriately classified and described for each research field. I believe that this review paper can provide useful information to many readers on climate change and groundwater research, and therefore deserves publication in this journal 'Sustainability'. 

Reviewer 3 Report

Thanks, please find attached my comments

good luck

Reviewer 4 Report

Brief summary

The paper, well written and organized, is a very clear and comprehensive summary of the latest investigation approaches to climate change impacts on the groundwater availability. The paper starts from the well-documented consideration that the main climate change impacts are on the groundwater recharge process. In the main part of the work an excellent analysis of different methods of groundwater recharge evaluation is presented, focusing on two important issues inherent the hydrological modelling approaches: the quantification and the reduction of uncertainty and the choice of appropriate downscaling methods in the GCM model applications. The conclusion summarizes the main remarks with a good critical approach.

Broad comments

The main suggestion I would point out is to modify the title. The paper is a review of different methods of estimation of GW recharge under climate change. In this review the scientific results of the literature papers are not analysed and discussed, the estimation of the impacts on the groundwater of climate change are not described or examinate; rather it is given greater weight to the estimation methods used in the recent literature. Thus, I believe it is righter to change the title in something like “Methos of GW recharge estimation under climate change: A review”.

Also, I suggest to insert in the Materials and Methods chapter a sentence or two explaining the evaluation criteria of the relevance of the studies. You should better explain on what basis you chose 50 manuscripts and eliminated the others (line 65-68).

Specific comments

Line 113 – 114 Check that the reference number (12) is right.  In the References, the corresponding title mentions Australia, but in text you talk about Russia. This reference is repeated in the Line 119, where you talk about Australia.

Figure 3: reverse the order of the years for easy reading of the graph.

Round 2

Reviewer 1 Report

The comments in the first round have not been addressed  in the revised manuscript.

Round 3

Reviewer 1 Report

The revised manuscript can be accepted in current version.